# Phylogeny of the plant genus *Pachypodium* (Apocynaceae)

Dylan O. Burge[1], Kaila Mugford[2], Amy P. Hastings[3] and
Anurag A. Agrawal[3]

[1] Department of Botany, University of British Columbia, Vancouver, British Columbia, Canada
[2] Department of Neurosciences, University of Toledo, Toledo, Ohio, USA
[3] Department of Ecology and Evolutionary Biology, Cornell University, Ithaca, New York, USA

## ABSTRACT

**Background.** The genus *Pachypodium* contains 21 species of succulent, generally spinescent shrubs and trees found in southern Africa and Madagascar. *Pachypodium* has diversified mostly into arid and semi-arid habitats of Madagascar, and has been cited as an example of a plant group that links the highly diverse arid-adapted floras of Africa and Madagascar. However, a lack of knowledge about phylogenetic relationships within the genus has prevented testing of this and other hypotheses about the group.

**Methodology/Principal Findings.** We use DNA sequence data from the nuclear ribosomal ITS and chloroplast *trn*L-F region for all 21 *Pachypodium* species to reconstruct evolutionary relationships within the genus. We compare phylogenetic results to previous taxonomic classifications and geography. Results support three infrageneric taxa from the most recent classification of *Pachypodium*, and suggest that a group of African species (*P. namaquanum*, *P. succulentum* and *P. bispinosum*) may deserve taxonomic recognition as an infrageneric taxon. However, our results do not resolve relationships among major African and Malagasy lineages of the genus.

**Conclusions/Significance.** We present the first molecular phylogenetic analysis of *Pachypodium*. Our work has revealed five distinct lineages, most of which correspond to groups recognized in past taxonomic classifications. Our work also suggests that there is a complex biogeographic relationship between *Pachypodium* of Africa and Madagascar.

## INTRODUCTION

*Pachypodium* (Apocynaceae) comprises 21 species of spinescent, succulent, xerophytic shrubs and small trees distributed in Madagascar and southern Africa (Table 1). *Pachypodium* is well known for its diverse array of growth forms, from dwarf shrubs to tall monopodial 'bottle trees', as well as its showy insect-pollinated flowers (Fig. 1; Table 1; *Vorster & Vorster, 1973*; *Rauh, 1985*; *Lavranos & Röösli, 1996*; *Lavranos & Röösli, 1999*; *Rapanarivo et al., 1999*; *Lüthy, 2004*). The center of diversity for *Pachypodium* is Madagascar, with 16 endemic species; the remaining five species are restricted to

Corresponding author
Dylan O. Burge,
dylan.o.burge@gmail.com

**Table 1** *Pachypodium* species, sampling, geography, and traits.

| Taxon | Sampled | Geography | Form | Corolla |
|---|---|---|---|---|
| *Pachypodium ambongense* H.Poiss. | 1 | Madagascar | Shrub | White |
| *P. baronii* Constantin and Bois | 2 | Madagascar | Shrub | Red |
| *P. bispinosum* (L.f.) A.DC. | 1 | Southern Africa | Shrub | Pink |
| *P. brevicaule* Baker subsp. *brevicaule* | 3 | Madagascar | Shrub | Yellow |
| *P. brevicaule* Baker subsp. *leucoxanthum* Lüthy | 1 | Madagascar | Shrub | White |
| *P. decaryi* H.Poiss. | 3 | Madagascar | Shrub | White |
| *P. densiflorum* Baker | 8 | Madagascar | Shrub | Yellow |
| *P. eburneum* Lavranos and Rapan. | 2 | Madagascar | Shrub | White |
| *P. geayi* Costantin and Bois | 1 | Madagascar | Tree | White |
| *P. horombense* H.Poiss. | 3 | Madagascar | Shrub | Yellow |
| *P. inopinatum* Lavranos | 1 | Madagascar | Shrub | White |
| *P. lamerei* Drake | 7 | Madagascar | Tree | White |
| *P. lealii* Welw. | 1 | Southern Africa | Tree | White |
| *P. menabeum* Leandri | 3 | Madagascar | Tree | White |
| *P. mikea* Lüthy | 1 | Madagascar | Tree | White |
| *P. namaquanum* (Wyley ex Harv.) Welw. | 1 | Southern Africa | Shrub | Red |
| *P. rosulatum* Baker subsp. *bemarahense* Lüthy and Lavranos | 1 | Madagascar | Shrub | Yellow |
| *P. rosulatum* Baker subsp. *bicolor* (Lavranos and Rapan.) Lüthy | 1 | Madagascar | Shrub | Yellow |
| *P. rosulatum* Baker subsp. *cactipes* (K.Schum.) Lüthy | 1 | Madagascar | Shrub | Yellow |
| *P. rosulatum* Baker subsp. *gracilius* (H.Perrier) Lüthy | 2 | Madagascar | Shrub | Yellow |
| *P. rosulatum* Baker subsp. *makayense* (Lavranos) Lüthy | 1 | Madagascar | Shrub | Yellow |
| *P. rosulatum* Baker subsp. *rosulatum* | 5 | Madagascar | Shrub | Yellow |
| *P. rutenbergianum* Vatke | 1 | Madagascar | Tree | White |
| *P. saundersii* N.E.Br. | 1 | Southern Africa | Shrub | White |
| *P. sofiense* (H.Poiss.) H.Perrier | 1 | Madagascar | Tree | White |
| *P. succulentum* (L.f.) A.DC. | 1 | Southern Africa | Shrub | Pink |
| *P. windsorii* H. Poiss. | 2 | Madagascar | Shrub | Red |

**Notes.**

*Taxon*, according to revision of *Lüthy (2004)*; *Sampled*, number of individuals sampled for genetic analysis; *Geography*, indicates whether the species is endemic to Madagascar or southern Africa; *Corolla*, indicates the overall color of the corolla (*Rapanarivo et al., 1999*; *Lüthy, 2006*).

southern Africa (Fig. 1; Table 1). Most *Pachypodium* species are narrowly distributed, with specialized ecology (*Vorster & Vorster, 1973*; *Lüthy, 2004*; *Rapanarivo et al., 1999*); habitats vary from desert to subhumid grassland, although most species are restricted to extremely arid environments (i.e., 8–34 cm annual precipitation; *Rapanarivo et al., 1999*). Those species that occur in more mesic habitats (up to 200 cm annual precipitation; *Rapanarivo et al., 1999*) tend to occupy rocky outcrops that are probably edaphically arid.

The showy flowers and unusual growth forms of *Pachypodium* have made them a favorite of horticulturists, leading to the exploitation of wild plants (*Lüthy, 2006*). Over-collecting combined with habitat destruction (*Goodman & Benstead, 2003*) has led to international trade restrictions, highlighting the need for improved systematic understanding of *Pachypodium*.

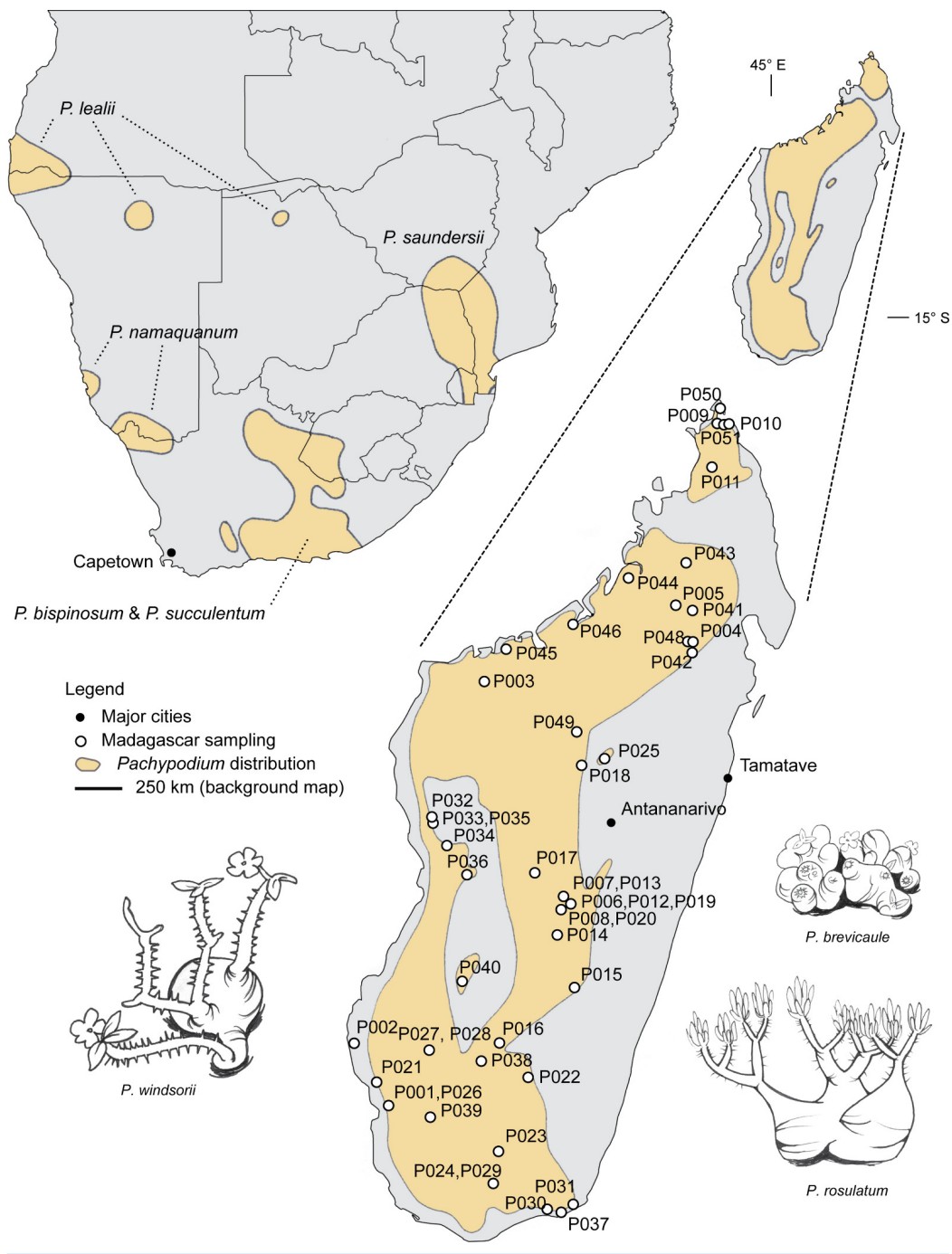

**Figure 1 Geographic distribution of *Pachypodium*.** Inset is sampling of *Pachypodium* in Madagascar (Appendix 1). Sampling in Africa not mapped. Data for distributions is approximate, adapted from *Lüthy (2006)* and *Vorster & Vorster (1973)*.

In Madagascar, *Pachypodium* forms a component of the strongly endemic xerophytic flora (*Rapanarivo et al., 1999*). These high levels of endemism in the xerophytic flora of Madagascar are attributed to the great antiquity of arid conditions on the island

(*Koechlin, 1972*); a climate suitable for the growth of xerophytic plants is thought to have prevailed in at least part of Madagascar throughout the Cenozoic (0–65 Ma; *Wells, 2003*). In addition, *Pachypodium* is part of a large group of arid-adapted plants—including many other succulents, such as *Euphorbia* and *Aloe*—with representatives in both Africa and Madagascar (*Leroy, 1978*; *Jürgens, 1997*); these plants provide evidence for a biogeographic link between arid regions of Africa and Madagascar, many of which are widely disjunct from one-another or isolated by intervening mesic habitats (*Leroy, 1978*). However, without an explicit phylogenetic framework, it is impossible to decipher the history of *Pachypodium* diversification in the Afro-Malagasy region.

Several taxonomic classifications of Malagasy *Pachypodium* have been proposed on the basis of morphological characteristics (Table 2). However, the African species of *Pachypodium* have been inconsistently treated, leading to a lack of knowledge on their relationship to Malagasy species. Some workers have assumed that the long temporal and wide geographic separation between Madagascar and Africa (*Yoder & Nowak, 2006*) corresponds to a deep genetic divergence between *Pachypodium* species from the two regions (*Perrier de la Bâthie, 1934*; *Lüthy, 2004*). Indeed, *Perrier de la Bâthie (1934)* suggested that the two groups might not be one-another's closest relatives. Nevertheless, the implied divergence is not strongly reflected by morphology; *Lüthy (2004)* cited only one trait—the presence of brachyblasts in African species—to separate the two groups. Overall, the monophyly of African and Malagasy *Pachypodium*, proposed infrageneric taxa, and *Pachypodium* itself, has never been tested.

We reconstruct the evolutionary history of *Pachypodium* using nuclear ribosomal ITS and chloroplast *trn*L-F DNA sequence data. Two additional chloroplast loci were included in the project design (*trn*S-G intergenic spacer and *rp*L16; *Shaw et al., 2005*), but proved insufficiently variable to justify further development. However, both ITS and *trn*L-F have proven utility for species-level phylogenetic reconstruction in plants (*Baldwin et al., 1995*; *Shaw et al., 2005*). Our specific aims were to (1) test infrageneric classifications of *Pachypodium* and (2) determine relationships between the African and Malagasy members of *Pachypodium*, including patterns of diversification between the two landmasses.

## MATERIALS AND METHODS

### Taxon sampling

We generated new ITS and *trn*L-F sequences from 56 *Pachypodium* samples representing all 27 minimum-rank taxa (species and subspecies) in the most recent revision of the genus (*Lüthy, 2004*; Tables 1 and 2). An additional ITS sequence was generated for *Funtumia africana*—a close relative of *Pachypodium* (*Livshultz et al., 2007*)—for use in rooting the ITS tree. *Pachypodium* and *Funtumia* tissues for DNA analysis were taken from greenhouse or garden plants (Appendix 1). Tissues were obtained by D. Burge, Walter Röösli, Nicholas Plummer, and Anurag Agrawal. Specimens were identified by W. Röösli, N. Plummer, or D. Burge according to the taxonomic revision of *Lüthy (2004)* and subsequent descriptions of new taxa (*Lüthy, 2005*; *Lüthy & Lavranos, 2005*). Plants were selected based on geographic distribution, with a larger amount of sampling for widespread taxa.

**Table 2** Summary of *Pachypodium* classification.

| Subgenus | Section | Series | Species or subspecies |
|---|---|---|---|
| *Nesopodium* | *Gymnopus* | *Ramosa* | *P. brevicaule* subsp. *brevicaule* |
| | | | *P. brevicaule* subsp. *leucoxanthum* |
| | | | *P. rosulatum* subsp. *bemarahense* |
| | | | *P. rosulatum* subsp. *bicolor* |
| | | | *P. rosulatum* subsp. *cactipes* |
| | | | *P. rosulatum* subsp. *gracilius* |
| | | | *P. rosulatum* subsp. *makayense* |
| | | | *P. rosulatum* subsp. *rosulatum* |
| | | *Densiflora* | *P. densiflorum* |
| | | | *P. eburneum* |
| | | | *P. horombense* |
| | | | *P. inopinatum* |
| | *Leucopodium* | *Contorta* | *P. decaryi* |
| | | | *P. rutenbergianum* |
| | | | *P. sofiense* |
| | | *Ternata* | *P. geayi* |
| | | | *P. lamerei* |
| | | | *P. mikea* |
| | | *Pseudoternata* | *P. ambongense* |
| | | | *P. menabeum* |
| | *Porphyropodium* | | *P. baronii* |
| | | | *P. windsorii* |
| *Pachypodium* | | | *P. bispinosum* |
| | | | *P. lealii* |
| | | | *P. namaquanum* |
| | | | *P. saundersii* |
| | | | *P. succulentum* |

**Notes.**

See Table 1 for taxon authorities; table includes later descriptions of new *Pachypodium* species by *Lüthy* (*2005*; *P. mikea*), *Lüthy & Lavranos* (*2005*; *P. rosulatum* subsp. *bemarahense*), and *Lüthy* (*2008*; *P. brevicaule* subsp. *leucoxanthum*).

Between one and eight populations of each taxon were used (Tables 1 and 2). Additional non-*Pachypodium trn*L-F sequences, for rooting trees, were obtained from GenBank (*F. africana* [EF456206], *Holarrhena curtisii* [EF456122], *Kibatalia macrophylla* [EF456119], *Malouetia bequaertiana* [EF456243], and *Mascarenhasia lisianthiflora* [EF456174]). These taxa were selected on the basis of their close relationship with *Pachypodium* (*Livshultz et al., 2007*).

## Molecular methods

Total genomic DNA was extracted from silica-dried leaves or seeds using the DNeasy Plant Mini Kit (Qiagen, Germantown, MD) according to the manufacturer's instructions. For seeds, up to three excised embryos from a single parent plant were pooled prior to DNA extraction (*Burge & Barker, 2010*). DNA was extracted from seeds when silica-dried material for the same plant was not available, or proved recalcitrant to extraction of

high quality DNA. All polymerase chain reactions were performed using Qiagen *Taq* DNA Polymerase. Amplifications were performed using an initial incubation at 94°C for 10 min and 30 cycles of three-step PCR (1 min at 94°C, 30 s at 45°C, and 2 min at 72°C), followed by final extension at 72°C for 7 min. PCR was performed on a Perkin Elmer GeneAmp thermocycler. The primers ITS4 (*White et al., 1990*) and ITSA (*Blattner & Kadereit, 1999*) were used to amplify the ITS1-5.8S-ITS2 region of the nuclear ribosomal DNA. Primers 'c' and 'f' (*Taberlet et al., 1991*) or a combination of these with internal primers 'd' and 'e' were used to amplify the *trn*L-F chloroplast region. For some plants, sequencing of ITS was problematic as a result of variation in length among ITS copies present in individual plants. Consequently, cloning of the ITS region was required for some plants. Cloning was carried out using the pGEM-T Easy Vector kit (ProMega, Madison, WI) according to the manufacturer's instructions. NIA inserts were amplified directly from up to four positive colonies using the PCR protocol described above. For all PCR reactions, excess primer and dNTPs were removed using exonuclease I (New England Biolabs, Ipswich, MA [NEB]; 0.2 units/μl PCR product) and antarctic phosphatase (NEB; 1.0 unit/ μl PCR product) incubated for 15 min at 37°C followed by 15 min at 80°C. For sequencing we used Big Dye chemistry (Applied Biosystems, Foster City, CA) according to the manufacturer's instructions. Sequences were determined bidirectionally on an Applied Biosystems 3100 Genetic Analyzer at the Duke University Institute for Genome Science and Policy Sequencing Core Facility.

## Sequence editing and alignment

All sequences were assembled and edited in Sequencher 4.1 (Gene Codes Corporation). In the case of the five plants for which ITS was cloned, we assessed sequence variation using an alignment of cloned sequences (hereafter 'isolates'). Two plants yielded pools of identical isolates (P011 and P021, Appendix 1), one yielded four different types of isolate (P053), and two were represented by a single successfully cloned isolate (P046 and P048). For the plant with more than one isolate type (P053), we included all four isolates in the phylogenetic analyses of ITS; for the plants with identical isolates, we selected a single isolate to represent each plant. New ITS and *trn*L-F sequences for *Pachypodium* were deposited in GenBank (Appendix 1).

The 60 new ITS and 55 new *trn*L-F sequences, along with additional outgroup sequences from GenBank, were used to create separate alignments for the two regions (Table 3; Alignments S2 and S3). Sequences were aligned in MUSCLE (*Edgar, 2004*) under default settings. For ITS, several indel- and repeat-rich regions (54 bp total) were excluded due to alignment ambiguity. A portion of *trn*L-F not available for some taxa (the 3′ *trn*L intron) was recoded as missing data. Indels were not recoded for analysis.

Following individual alignment of ITS and *trn*L-F, we endeavored to create a combined alignment. Preliminary analyses showed that for the single *Pachypodium* sample represented by more than one cloned ITS sequence (P053; Table 1), the four sequences formed a monophyletic group. Thus, a single sequence from this group was selected at random. For the final combined alignment (Alignment S2), the entire *trn*L-F region was coded

**Peer**J

**Table 3** Summary statistics for DNA alignments.

| Name | Region | Terminals | Total length | Included length | G + C | Variable | PIC |
|------|--------|-----------|--------------|-----------------|-------|----------|-----|
| Alignment S1 | ITS | 60 | 658 | 604 | 53.7% | 156 (226) | 110 (116) |
| Alignment S2 | *trn*L-F | 59 | 961 | 961 | 36.4% | 33 (64) | 18 (36) |
| Alignment S3 | ITS and *trn*L-F | 61 | 1619 | 1565 | 43.1% | 184 (285) | 114 (140) |

Notes.

*Total Length*, the length of the complete alignment, counting portions excluded from analysis; *Included length*, the total number of characters included in the phylogenetic analysis. *G + C*, the G + C content of the complete (total length) alignment; *Variable*, the number of variable characters in the ingroup, followed by the number of variable characters in the full alignment (in parentheses); *PIC*, the number of parsimony-informative characters in the ingroup, followed by the number of parsimony informative characters in the full alignment (in parentheses).

as missing data for the two *Pachypodium* samples from which *trn*L-F was not obtained (*P. bispinosum* A049 and *P. brevicaule* subsp. *leucoxanthum*, P066). To test for conflict between the nuclear (ITS) and chloroplast (*trn*L-F) portions of the alignment, we used the incongruence length difference test (*Farris et al., 1995*), implemented in PAUP* v 4.0 (*Swofford, 2002*) as the partition homogeneity test. The test used 1000 random repetitions of the parsimony analysis described below (see Phylogenetic analyses). Results showed significant disagreement between ITS and *trn*L-F ($P = 0.047$; 953/1000 trees). To account for this conflict, we ran all of our phylogenetic analyses on the separate *trn*L-F and ITS alignments, noting any well-supported conflicts between the results, and compared these to results from the combined alignment (see Discussion).

## Phylogenetic analyses

Trees were reconstructed using Bayesian, maximum likelihood (ML), and maximum parsimony (MP) techniques. Bayesian analyses were carried out based on the best fit model of evolution from jModelTest 2, under default parameters (*Posada & Crandall, 1998*; *Guindon & Gascuel, 2003*; *Darriba et al., 2012*; ITS: GTR + I + G; *trn*L-F: GTR + I). Bayesian sampling was performed in MrBayes v 3.2.1 (*Ronquist & Huelsenbeck, 2003*), using the models of sequence evolution identified by jModelTest 2; all other parameters of MrBayes were left at default values; for the combined tree, no rate or model constraints were imposed between the two partitions. Analyses were carried out as follows: (1) three separate runs of $1 \times 10^7$ MCMC generations, sampling every 1000 generations, (2) examination of run output for convergence (standard deviation of split frequencies nearing 0.001) (3) removal of the first 1000 samples (10%) as burnin after visual inspection of likelihood score plots, (4) comparison of consensus trees for each run, and (5) combination of post-burnin samples from all three runs to compute a 50% majority-rule consensus tree (conducted in PAUP* v 4.0 (*Swofford, 2002*)). A partitioned model of sequence evolution was used for the analysis of the combined data.

Maximum likelihood analyses were carried out in GARLI v 2.0 (*Zwickl, 2006*). For each alignment, two search replicates were performed in a single execution. Models of evolution were the same as those described for Bayesian analyses, with a partitioned model applied to the combined alignment. Other parameters were kept at default. Statistical support was inferred with 100 replicates of bootstrap reweighting (*Felsenstein, 1985*), implemented as in the tree search.

Maximum parsimony analysis was conducted in PAUP* v 4.0 (*Swofford, 2002*). An initial heuristic search of 100 random taxon addition replicates was conducted with tree-bisection-reconnection branch swapping (TBR) and MULPARS in effect, retaining only ten trees from each replicate. A strict consensus of these trees was then used as a constraint tree in a second heuristic search using the similar parameters as above, but with 1000 random sequence addition replicates, and retaining 100 trees per addition replicate. We used this method due to the excessive number of trees generated by unconstrained searches. This strategy checks for shorter trees than those found by the initial search, demonstrating that the final consensus tree reflects all of the most parsimonious trees (*Catalán, Kellogg & Olmstead, 1997*). We also ran searches on the three alignments using an unconstrained search with the nearest neighbor interchange (NNI) swapping algorithm, which produced trees of exactly the same length as the constrained searches. In the interest of brevity, we present results only for the constrained searches. We estimated Bootstrap support (*Felsenstein, 1985*) for our parsimony trees using 100 pseudoreplicates and the same search setting as described above, including use of a constraint tree. We treated gaps as missing data for all phylogenetic analyses.

### Topology testing

We used Templeton's nonparametric test (1983), as implemented in PAUP* v 4.0 (*Swofford, 2002*), to evaluate several key phylogenetic relationships. Templeton's test compares pairs of topologies, measuring relative statistical support for the trees within a sequence dataset (alignment). For these tests, we compared the best tree from the original parsimony tree search (see above) to the best tree from a search using a constraint (e.g., African *Pachypodium* constrained as monophyletic). For more on these tests, see below (Results).

## RESULTS

### Alignments

The ITS region had an aligned length of 658 bp (Alignment S1). Of the 156 (included) variable positions within the ingroup, 110 were parsimony informative (Table 3). The *trn*L-F region had an aligned length of 961 bp (Alignment S2). Of the 33 variable positions within the ingroup, 18 were parsimony-informative (Table 3). The combined alignment contained 61 terminals, with an aligned length of 1619 bp (Alignment S3). Of the 184 (included) variable positions in the ingroup, 114 were parsimony informative.

### Phylogenetic trees

The Bayesian 50% majority-rule consensus tree for ITS contained 13 internal nodes with a posterior probability (PP) of 1.0 (Treefile S4A; Fig. 2). By contrast, the *trn*L-F-based Bayesian tree contained only five internal nodes with a PP of 1.0 (Treefile S5A; Fig. 2). The combined ITS and *trn*L-F tree contained 17 internal nodes with a PP of 1.0 (Treefile S6A; Fig. 3).

Maximum parsimony searches based on ITS data alone resulted in 4851 trees of 324 steps (Table 4; Treefile S4B); a total of 12 internal nodes had bootstrap (BS) support greater than or equal to 95% (Treefile S4C). Searches using *trn*L-F data alone resulted in 8 trees

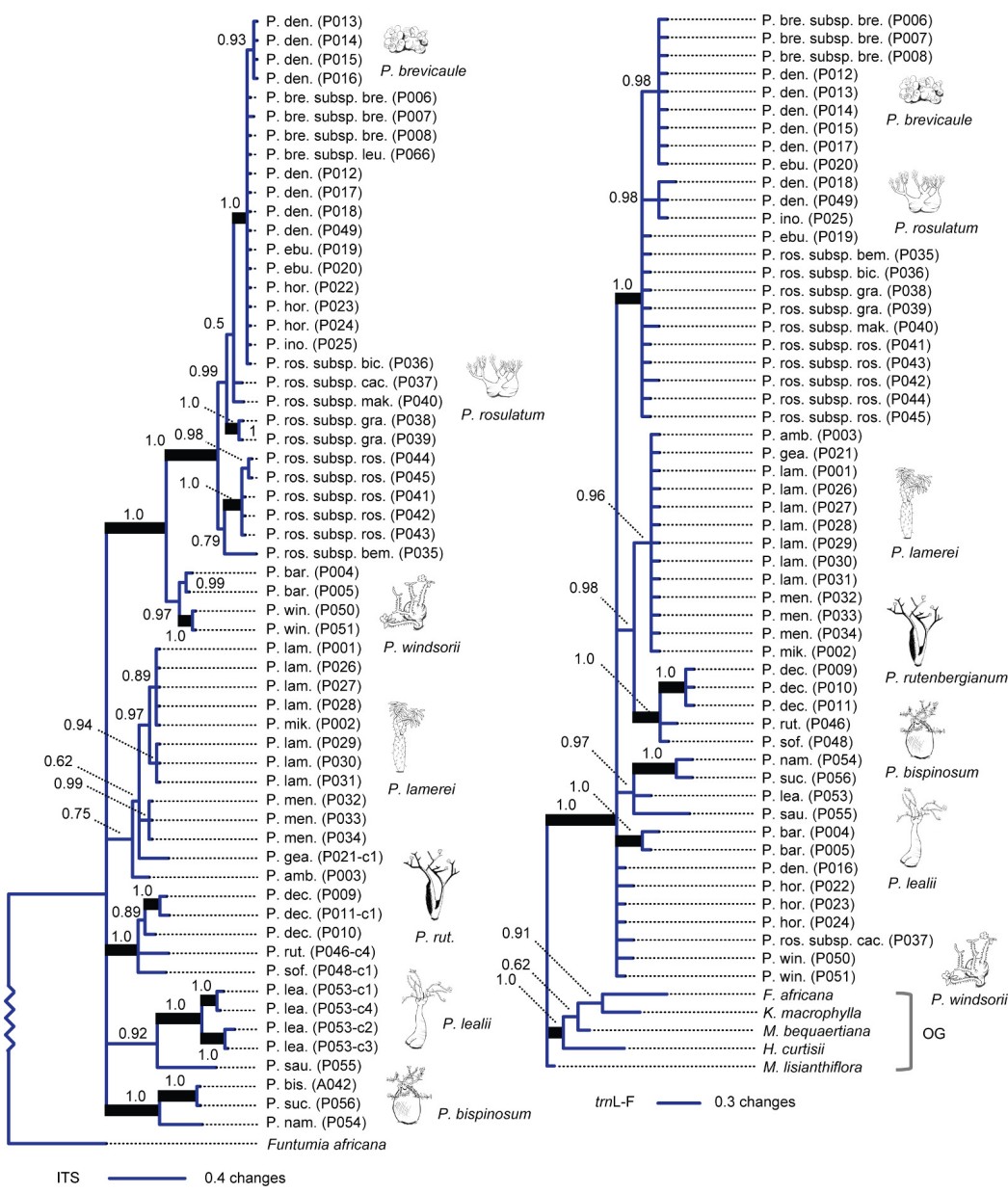

**Figure 2 Bayesian consensus phylograms for individual genetic regions.** Left, ITS; right, *trn*L-F. Numbers above branches are Bayesian posterior probability (PP) from the 50% majority rule consensus tree; thickened branches have PP of 1.0. Taxon names are abbreviated (see Table 1). ITS tree is midpoint rooted. Zigzag line indicates that the branch connecting the outgroup to *Pachypodium* is not shown to scale (see Treefiles S4 and S5).

of 71 steps (Table 4; Treefile S5B); only one internal node had BS support greater than or equal to 95% (Treefile S5C). Searches on the combined ITS and *trn*L-F data resulted in 4582 trees of 394 steps (Table 4; Treefile S6B); a total of 8 internal nodes had BS support greater than or equal to 95% (Treefile S6C; Fig. 3). In all cases, use of a constraint tree failed to find any trees of equal or shorter length that contradicted the respective consensus trees.

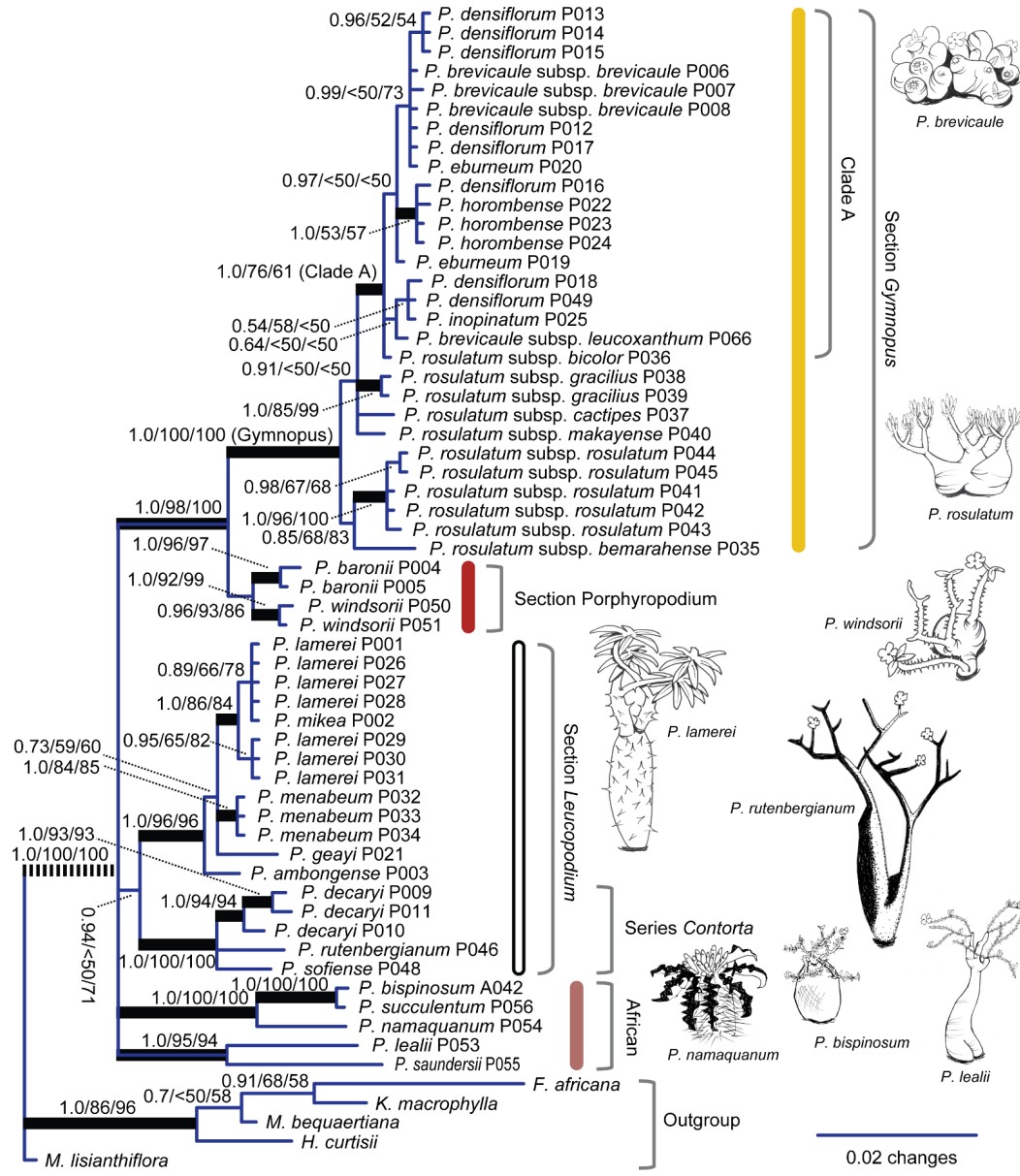

**Figure 3 Bayesian consensus phylogram for combined data.** Numbers above branches are (from left to right) (1) Bayesian posterior probability (PP) from the 50% majority rule consensus tree, (2) maximum parsimony bootstrap support, and (3) maximum likelihood bootstrap support; thickened branches have PP of 1.0. Selected subgeneric taxa are from of *Lüthy (2004)*; colored bars indicate predominant color of corolla lobes (Table 1). Dashed line indicates a branch not shown to scale (see Treefile S6).

Maximum likelihood (ML) analyses support similar relationships to those indicated by maximum parsimony and Bayesian analyses. The best ML tree for ITS alone contained 14 internal nodes with BS support greater than or equal to 95% (Treefiles S4D and S4E). By contrast, the *trn*L-F-based ML tree contained only one internal node with BS greater than or equal to 95% (Treefiles S5D and S5E). The best ML tree based on ITS combined with

**Table 4  Summary statistics for maximum parsimony tree searches.**

| Tree | Region | Total MP trees | Steps | CI | RI |
|------|--------|----------------|-------|------|------|
| Treefile S4B | ITS | 4851 | 324 | 0.82 | 0.95 |
| Treefile S5B | *trn*L-F | 8 | 71 | 0.93 | 0.97 |
| Treefile S6B | ITS and *trn*L-F | 4582 | 394 | 0.83 | 0.92 |

**Notes.**
CI, consistency index; RI, retention index.

*trn*L-F contained 10 internal nodes with BS support greater than or equal to 95% (Treefiles S6D and S6E; Fig. 3).

*Pachypodium* is recovered as monophyletic in the *trn*L-F tree (Fig. 2A), but lack of broad outgroup sampling for ITS prevents assessment of *Pachypodium* monophyly based on nuclear DNA; support for *Pachypodium* monophyly in the combined tree is driven by *trn*L-F. Six of the 11 minimum-rank *Pachypodium* taxa (species and subspecies) represented by more than one sampled plant (Table 1) are monophyletic in the combined tree, four with strong support (PP 1.0; MP bootstrap ≥ 95%; *P. baronii*, *P. decaryi*, *P. rosulatum* subsp. *rosulatum*, and *P. windsorii*; Fig. 3). The following multi-taxon clades are also recovered with high levels of support in the combined tree (PP = 1.0; MP BS ≥ 95%): (1) the Malagasy *P. decaryi*, *P. rutenbergianum*, and *P. sofiense*, (2) the African *P. lealii* and *P. saundersii*, (3) the African *P. namaquanum*, *P. succulentum*, and *P. bispinosum*, (4) an 11-taxon group corresponding to section *Gymnopus* (Table 2), and (5) a smaller group nested within *Gymnopus* comprising *P. brevicaule* subsp. *brevicaule*, *P. densiflorum*, *P. eburneum*, *P. inopinatum*, and *P. rosulatum* subsp. *bicolor*.

## Topology test results

Based on the results from our initial tree searches (Figs. 2 and 3), we were interested to know whether the data could reject (1) monophyly of African *Pachypodium*, (2) monophyly of Malagasy *Pachypodium*, and (3) reciprocal monophyly of African and Malagasy *Pachypodium*. These tests were done by comparing the most parsimonious tree from the original heuristic tree search to the most parsimonious tree from a search in which one of the above groups was used as a constraint. We carried out these analyses for ITS and for the combined data. Because the *trn*L-F region was not sampled for one of the African species (*P. bispinosum*), we were not able to evaluate these hypotheses on the basis of chloroplast DNA alone. For ITS, the shortest tree compatible with the first constraint (monophyletic African *Pachypodium*) was four steps longer (328 steps) than the unconstrained tree (324 steps), which was judged not to be significant based on a Templeton test ($P = 0.25$). A similar result was obtained for the combined data (396 steps in the constrained tree versus 394 steps in the unconstrained tree; $P = 0.64$). For the second constraint (monophyletic Malagasy *Pachypodium*), the shortest ITS tree compatible with the constraint was only one step longer than the unconstrained tree, which was also not significant based on a Templeton test ($P = 0.71$); again, the combined data were in agreement (both trees 394 steps; $P = 1.0$). Finally, for the third constraint (reciprocal monophyly of African and Malagasy *Pachypodium*), the shortest ITS tree compatible with

the constraint was five steps longer than the unconstrained tree, which was not a significant difference ($P = 0.1$); the combined data support this result 398 steps in the constrained tree versus 394 steps in the unconstrained tree; $P = 0.29$.

## DISCUSSION

### Conflict

Our study identified significant conflict between the nuclear and chloroplast datasets, based on the incongruence length difference test (see Materials and Methods). However, we elected to combine the datasets for further analysis. Our choice to unite the conflicting datasets is a conditional combination approach (*Bull et al., 1993*; *Huelsenbeck, Bull & Cunningham, 1996*), based on the lack of conflict between well-supported internal nodes (also called "hard conflict") in the *trn*L-F and ITS trees (Fig. 2). Our combined approach should be treated as tentative, despite the lack of clearly conflicting internal nodes in ITS versus *trn*L-F trees.

### Phylogenetic relationships

Our *trn*L-F trees suggest that *Pachypodium* is monophyletic, based on sampling of closely related genera. However, because of a lack of appropriate outgroups for the nuclear region (ITS), we were unable to evaluate the hypothesis of *Pachypodium* monophyly on the basis of both genomes. Nevertheless, the monophyly of *Pachypodium* is generally uncontroversial, and is supported by other molecular phylogenetic research (*Livshultz et al., 2007*), as well as a suite of morphological characters, including alternate phyllotaxy (most Apocynaceae have opposite leaf arrangement), a horseshoe-shaped retinacle (the connection between the anther and the style head), loss of colleters associated with the calyx, and stem succulence (*Sennblad, Endress & Bremer, 1998*).

Overall, our data do not provide sufficient phylogenetic resolution to draw conclusions concerning the monophyly or non-monophyly of African and Malagasy *Pachypodium*. Despite the recovery of several well-supported lineages in both African and Malagasy *Pachypodium*, the basal branching relationships among these lineages is not well resolved by ITS, *trn*L-F, or the combined data (Figs. 2 and 3). However, it should be noted that *trn*L-F provides some evidence for the cohesiveness of African *Pachypodium* (Fig. 2B); lack of ITS data for reliably vouchered *P. bispinosum* makes it impossible to test this hypothesis using *trn*L-F, although sequence data for samples of *P. bispinosum* of unknown wild origin (horticultural strains) do group with other African species in *trn*L-F trees (D. Burge and A. Agrawal, unpublished data). In general, there are four mutually exclusive hypotheses on the relationship between African and Malagasy *Pachypodium*, each of which may represent a valid interpretation of our results: (1) reciprocally monophyletic African and Malagasy *Pachypodium*, (2) monophyletic Malagasy *Pachypodium* derived from within a basal grade of African *Pachypodium*, rendering African *Pachypodium* paraphyletic, (3) monophyletic African *Pachypodium* arising from a basal grade of Malagasy *Pachypodium*, with Malagasy *Pachypodium* paraphyletic, and (4) neither African nor Malagasy *Pachypodium* monophyletic. Topology tests could not reject any of these hypotheses.

A recent estimate of 37–64 Ma for the divergence of stem Apocynaceae from its closest relatives among the Gentianales (*Bell, Soltis & Soltis, 2010*) implies that the crown age of *Pachypodium* is probably more recent than the ~80 Ma timing for the isolation of Madagascar from Africa (*Yoder & Nowak, 2006*). In fact, a recent review of Madagascar biogeography suggests that most of Madagascar's biotic connections are best explained by long-distance dispersal during the Cenozoic, rather than ancient Gondwanan vicariance (*Yoder & Nowak, 2006*). Thus, if *Pachypodium* did not originate in Madagascar, it must have arrived on the island via long-distance dispersal. However, the lack of phylogenetic resolution among major African and Malagasy lineages of *Pachypodium* prevents preventing reliable reconstruction of geographic range evolution, including dispersal-vicariance scenarios between Africa and Madagascar.

Additional molecular phylogenetic work will be required to obtain better support for basal-branching relationships in *Pachypodium*, particularly the relationship between African and Malagasy species. This work will likely require the sequencing of additional loci, from both the chloroplast and nuclear genome. Resolution of relationships among species from *Lüthy*'s (*2004*) section *Gymnopus* will also require additional work. In *Gymnopus*, a number of widespread species (e.g., *P. densiflorum* and *P. brevicaule*) are non-monophyletic. The lack of phylogenetic cohesiveness among populations in such species is consistent with both hybridization following initial divergence, as well as incomplete lineage sorting (retention of ancestral polymorphisms; *Pamilo & Nei, 1988*; *Maddison & Knowles, 2006*), a phenomenon that often occurs during rapid diversification). For future studies on section *Gymnopus*, rapidly evolving genetic markers such as low-copy nuclear genes may help to discern species-trees from gene-trees, while population genetic markers such as AFLPs and microsatellites might also help to decipher complex relationships, especially in regions of geographic overlap among species.

## Testing classification

Our exhaustive sampling of *Pachypodium* species and subspecies (Table 1) has provided the opportunity to test existing morphology-based hypotheses on infrageneric relationships. Our results support the most recent infrageneric classification of *Pachypodium* proposed by *Lüthy* (*2004*; Table 2). *Lüthy*'s (*2004*) shrubby, predominantly yellow-flowered section *Gymnopus* is clearly monophyletic (Fig. 3, PP 1.0; MP & ML BS 100%), as is the shrubby, red-flowered section *Porphyropodium* (Fig. 3, PP 0.98; MP and ML 98%). Our results also indicate a very close relationship between *Porphyropodium* and *Gymnopus* (Fig. 3, PP 0.96; MP and ML BS >86%), a relationship not emphasized by past classifications. Finally, the third section recognized in *Lüthy*'s (*2004*) classification, the mostly arborescent, white-flowered *Leucopodium*, is marginally supported in the combined phylogenetic tree (Fig. 3, PP 0.94; ML BS 71%). Overall, our results also support the tradition of using corolla color as a basis for circumscription of taxa within *Pachypodium* (Fig. 3; *Poisson, 1924*; *Pichon, 1949*; *Perrier de la Bâthie, 1934*; *Lüthy, 2004*). Nonetheless, we agree with *Lüthy (2004)* that an ideal infrageneric classification should use multiple morphological characteristics to define groups.

Below the section level, previous classifications of *Pachypodium* are not well supported by our molecular phylogenetic results. One clear exception is *Lüthy*'s (*2004*) series *Contorta* (Table 2), which was defined on the basis of seed morphology to include the arborescent *P. rutenbergianum* and *P. sofiense*, as well as the limestone-endemic *P. decaryi*. Our results show that this group is strongly monophyletic (Fig. 3, PP 1.0; MP and ML BS 100%), confirming the detailed work of *Lüthy (2004)*. However, this contrasts with most previous opinions. *Pichon (1949)*, for example, allied *P. decaryi* with another limestone endemic, *P. ambongense*.

Within section *Gymnopus*, *Lüthy*'s (*2004*) series *Densiflora* (Table 2) roughly corresponds to a clade that we recover nested inside *Gymnopus* (Fig. 3, Clade A, PP 1.0; MP BS 76%). However, Clade A includes *P. rosulatum* subsp. *bicolor* and *P. brevicaule* subsp. *brevicaule*, both considered members of series *Ramosa* by *Lüthy (2004)*. Our results indicate that the floral characters used by *Lüthy (2004)* and others to define groups within *Gymnopus* (Table 2) are homoplasious.

Most past classifications of *Pachypodium* have dealt in very sparse detail, if at all, with the distinctive and morphologically heterogeneous African members of the genus. As discussed above (see Phylogenetic relationships), our results suggest that African *Pachypodium* comprises two distinctive lineages, one containing the morphologically similar *P. lealii* and *P. saundersii* (*Rapanarivo et al., 1999*), and a second containing the bizarre monopodial tree *P. namaquanum* and the tuberous shrubs *P. bispinosum* and *P. succulentum*. The close relationship between *P. lealii* and *P. saundersii* (Fig. 3, PP 1.0; MP BS 95%) has been noted for some time, as indicated by a reduction to synonymy under *P. saundersii* that was undertaken by *Rowley (1973)*. The close relationship of *P. namaquanum* to *P. bispinosum* and *P. succulentum* was less expected (Fig. 3, PP 1.0; MP and ML BS 100%). *Vorster & Vorster (1973)* did propose a close relationship between *P. namaquanum* and *P. bispinosum* based on corolla shape. However, these authors also proposed that the asymmetrical flowers of *P. succulentum* linked this species to *P. lealii* and *P. saundersii* more than to *P. bispinosum*. Our results clearly show that *P. bispinosum* and *P. succulentum* are one another's closest relatives, sister to *P. namaquanum*.

## Conservation

Conservation planning for threatened flora and fauna must take into consideration the evolutionary potential of populations and taxa (*Forest et al., 2007*). Ignoring evolutionary potential will lead to losses of diversity that compromise the ability of these groups to adapt and survive in the long-term. In the case of *Pachypodium*, phylogenetic results presented here show that several species and groups of species are strongly divergent from other *Pachypodium* (e.g., *P. decaryi* and most African *Pachypodium*; Fig. 3). These groups represent important islands of phylogenetic diversity within *Pachypodium*, the loss of which would drastically reduce the overall diversity of the genus. Many members of the *Gymnopus* section of *Pachypodium*, by contrast, are very shallowly divergent based on our results (Fig. 3). The members of *Gymnopus* are adapted to a great variety of habitats, and therefore may contain much ecological diversity in terms of local adaptation (*Lüthy, 2004*).

However, in comparison to highly divergent taxa such as *P. decaryi*, each individual *Gymnopus* taxon represents a very small proportion of the total phylogenetic diversity of *Pachypodium*. In light of the always-limited resources available for conservation, an effort should be made to prioritize the protection of phylogenetically divergent lineages of *Pachypodium* as well as the overall genetic diversity of the genus. We recommend stronger conservation measures—including greater restrictions on the trade of wild-collected plants—for very narrowly distributed species having Bayesian PP of 1.0 in the combined ITS and *trn*L-F tree (Fig. 3). This includes the Malagasy *P. baronii*, *P. windsorii*, and *P. decaryi*. The highly divergent African species are not included in this list due to their relatively wide geographic distributions.

## ACKNOWLEDGEMENTS

For tissue samples we thank Ralph Hoffmann, Nicholas Plummer, Walter Röösli, and the National Botanic Garden of Belgium, with logistical assistance from Frank Van Caekenberghe. Comments on drafts and data interpretation were provided by Jonas Lüthy, Nicholas Plummer, and Katherine Zhukovsky. Sketches of *Pachypodium* were rendered by Bonnie McGill.

## Appendix 1

Sampled plants and DNA sequences. For each plant the within-study code is in brackets, followed by collector and collector number, herbarium or living collection for deposition of voucher specimen (in parentheses; ZSS indicates living collection of Sukkulenten-Sammlung Zürich), provenance, and GenBank numbers for ITS and *trn*L-F; Abbreviation 's.n.' indicates no collection number.

*Funtumia africana*—[OG1] National Botanic Garden of Belgium 19514728 (BR), cultivated Plant; ITS: KC189049.

*Pachypodium ambongense*—[P003] W. Röösli, R. Hoffman, & M. Grubenmann, s.n., collected 25.xi.1989 (P, ZSS), Namoroka, Madagascar; ITS: HQ847410; *trn*L-F: HQ847465. *P. baronii*—[P004] A. Razafindratsira, s.n., collected 3.i.1988 (ZSS), Befandriana Nord, Madagascar; ITS: HQ847411; *trn*L-F: HQ847466. [P005] W. Röösli & B. Rechberger, s.n., collected xii.1990 (ZSS), Mandritsara, Madagascar; ITS: HQ847412; *trn*L-F: HQ847467. *P. bispinosum*—[A049] A. Agrawal, s.n. (DUKE), cultivated plant; ITS: JN256214. *P. brevicaule* subsp. *brevicaule*—[P006] W. Röösli & R. Hoffman 92/98 (ZSS), Mount Ibity, Madagascar; ITS: HQ847414; *trn*L-F: HQ847469. [P007] W. Röösli & R. Hoffman 43/01 (Z), Ranomainty, Madagascar; ITS: HQ847415; *trn*L-F: HQ847470. [P008] J. Lüthy, s.n., collected 1.vi.2006 (ZSS), Andrembesoa, Madagascar; ITS: HQ847416; *trn*L-F: HQ847471. *P. brevicaule* subsp. *leucoxanthum*—[P066] J. Lüthy, s.n., collected 6.i.2006 (ZSS), undisclosed locality, Madagascar; ITS: KC189050. *P. decaryi*—[P009] W. Rauh 72255 (HEID), Montagne des Francais, Madagascar; ITS: HQ847417; *trn*L-F: HQ847472. [P010] W. Röösli & R. Hoffman 22/99 (ZSS), Montagne des Francais, Madagascar; ITS: HQ847418; *trn*L-F: HQ847473. [P011] W. Röösli & R. Hoffman 22/00 (ZSS), Ankarana, Madagascar; ITS: HQ847419; *trn*L-F: HQ847474. *P. densiflorum*—[P012] W. Röösli & R. Hoffman 01/94 (ZSS), Mount Ibity, Madagascar; ITS: HQ847420; *trn*L-F: HQ847475.

[P013] W. Röösli & R. Hoffman 42/01 (ZSS), Ranomainty, Madagascar; ITS: HQ847421; *trn*L-F: HQ847476. [P014] W. Röösli & R. Hoffman, s.n., collected 1.xii.1992 (ZSS), Ambatofinandrahana, Madagascar; ITS: HQ847422; *trn*L-F: HQ847477. [P015] W. Röösli & B. Rechberger, s.n., collected 20.i.1989 (ZSS), Fianarantsoa, Madagascar; ITS: HQ847423; *trn*L-F: HQ847478. [P016] W. Röösli & R. Hoffman 57/98 (K, P, WAG), Plateaux Horombe, Madagascar; ITS: HQ847424; *trn*L-F: HQ847479. [P017] W. Röösli & R. Hoffman 45/93 (ZSS), 107 km W Antsirabe, Madagascar; ITS: HQ847425; *trn*L-F: HQ847480. [P018] W. Röösli & R. Hoffman 31/03 (ZSS), Mahatsinjo, Madagascar; ITS: HQ847426; *trn*L-F: HQ847481. [P049] A. Razafindratsira, s.n., collected xii.2006 (ZSS), Ambodiriana, Madagascar; ITS: HQ847427; *trn*L-F: HQ847482. ***P. eburneum***—[P019] W. Röösli & R. Hoffman 01/96 (P, MO, TAN, WAG, ZSS), Mount Ibity, Madagascar; ITS: HQ847428; *trn*L-F: HQ847483. [P020] J. Lüthy, s.n., collected 1.vi.2006 (ZSS), Andrembesoa, Madagascar; ITS: HQ847429; *trn*L-F: HQ847484. ***P. geayi***—[P021] W. Röösli & R. Hoffman 29/04 (ZSS), Ifaty, Madagascar; ITS: HQ847430; *trn*L-F: HQ847485. ***P. horombense***—[P022] W. Röösli & B. Rechberger, s.n., collected 21.xii.1990 (ZSS), Betroka, Madagascar; ITS: HQ847431; *trn*L-F: HQ847486. [P023] W. Röösli & R. Hoffman 34/01 (ZSS), Beraketa, Madagascar; ITS: HQ847432; *trn*L-F: HQ847487. [P024] W. Röösli & R. Hoffman 73/96 (WAG), Andalatanosy, Madagascar; ITS: HQ847433; *trn*L-F: HQ847488. ***P. inopinatum***—[P025] W. Röösli & R. Hoffman 46/93 (P, TAN, HEID, WAG, ZSS), Manakana, Madagascar; ITS: HQ847434; *trn*L-F: HQ847489. ***P. lamerei***—[P001] W. Röösli & R. Hoffman 18/06 (ZSS), Fiherenana River, Madagascar; ITS: HQ847435; *trn*L-F: HQ847490. [P026] W. Röösli & R. Hoffman 20/02 (ZSS), Fiherenana River, Madagascar; ITS: HQ847436; *trn*L-F: HQ847491. [P027] W. Röösli & R. Hoffman, s.n., collected 26.i.1994 (WAG, ZSS), Ihosy, Madagascar; ITS: HQ847437; *trn*L-F: HQ847492. [P028] W. Röösli & R. Hoffman, s.n., collected 24.i.1994 (ZSS), Beraketa, Madagascar; ITS: HQ847438; *trn*L-F: HQ847493. [P029] W. Röösli & R. Hoffman 31/01 (WAG, ZSS), Andalatanosy, Madagascar; ITS: HQ847439; *trn*L-F: HQ847494. [P030] W. Röösli & R. Hoffman 19/01 (ZSS), Lac Anony, Madagascar; ITS: HQ847440; *trn*L-F: HQ847495. [P031] W. Röösli & R. Hoffman 79/96 (P, WAG, ZSS), Fort Dauphin, Madagascar; ITS: HQ847441; *trn*L-F: HQ847496. ***P. lealii***—[P053] Huntington Botanic Garden 85642 (DUKE), cultivated Plant; ITS: HQ847442; JN256217; JN256216; JN256215; *trn*L-F: HQ847497. ***P. menabeum***—[P032] W. Röösli & B. Rechberger, s.n., collected 10.xii.1991 (ZSS), Antsalova, Madagascar; ITS: HQ847443; *trn*L-F: HQ847498. [P033] W. Röösli & R. Hoffman 07/03 (ZSS), Antsalova, Madagascar; ITS: HQ847444; *trn*L-F: HQ847499. [P034] W. Röösli & R. Hoffman 03/02 (ZSS), Bekopaka, Madagascar; ITS: HQ847445; *trn*L-F: HQ847500. ***P. mikea***—[P002] W. Röösli & R. Hoffman 26/05 (P, TAN), South of Morombe, Madagascar; ITS: HQ847446; *trn*L-F: HQ847501. ***P. namaquanum***—[P054] J. Lüthy, s.n. (University of Bern Institute of Plant Sciences, *living collection), cultivated Plant; ITS: HQ847447; trnL-F: HQ847502.* ***P. rosulatum* subsp. *bemarahense***—[P035] W. Röösli & R. Hoffman 08/03 (TAN), Antsalova, Madagascar; ITS: HQ847448; *trn*L-F: HQ847503. ***P. rosulatum* subsp. *bicolor***—[P036] W. Röösli & R. Hoffman 42/93 (P, MO, TAN, WAG, ZSS), Berevo,

Madagascar; ITS: HQ847449; *trn*L-F: HQ847504. **P. rosulatum subsp. cactipes**—[P037] W. Röösli & R. Hoffman 77/96 (BR, K, MO, P, TAN, WAG, ZSS), Fort Dauphin, Madagascar; ITS: HQ847450; *trn*L-F: HQ847505. **P. rosulatum subsp. gracilius**—[P038] W. Röösli & R. Hoffman 36/01 (ZSS), Isalo, Madagascar; ITS: HQ847451; *trn*L-F: HQ847506. [P039] W. Röösli & R. Hoffman 42/05 (K, MO, WAG), Bezaha, Madagascar; ITS: HQ847452; *trn*L-F: HQ847507. **P. rosulatum subsp. makayense**—[P040] W. Röösli & R. Hoffman 08/02 (MO, P, TAN), Makay, Madagascar; ITS: HQ847453; *trn*L-F: HQ847508. **P. rosulatum subsp. rosulatum**—[P041] W. Röösli & R. Hoffman 26/96 (WAG, ZSS), Antsakabary, Madagascar; ITS: HQ847454; *trn*L-F: HQ847509. [P042] W. Röösli & R. Hoffman 21/95 (MO, P, WAG, ZSS), Mandritsara, Madagascar; ITS: HQ847455; *trn*L-F: HQ847510. [P043] A. Razafindratsira, s.n., collected 30.xii.1991 (ZSS), Bealanana, Madagascar; ITS: HQ847456; *trn*L-F: HQ847511. [P044] W. Röösli & R. Hoffman 29/95 (ZSS), Ananalava, Madagascar; ITS: HQ847457; *trn*L-F: HQ847512. [P045] W. Röösli & R. Hoffman 23/03 (ZSS), Benetsy, Madagascar; ITS: HQ847458; *trn*L-F: HQ847513. **P. rutenbergianum**—[P046] W. Röösli & R. Hoffman 19a/95 (ZSS), Anjohibe, Madagascar; ITS: HQ847459; *trn*L-F: HQ847514. **P. saundersii**—[P055] M. Lehmann, s.n. (plants grown by N. Plummer) (DUKE), Karongwe Game Reserve, South Africa; ITS: HQ847460; *trn*L-F: HQ847515. **P. sofiense**—[P048] W. Röösli & R. Hoffman 14/96 (P, WAG), Mandritsara, Madagascar; ITS: HQ847461; *trn*L-F: HQ847516. **P. succulentum**—[P056] J. Lavranos, s.n. (University of Bern Institute of Plant Sciences, living collection), Grahamstoon, South Africa; ITS: HQ847462; *trn*L-F: HQ847517. **P. windsorii**—[P050] A. Razafindratsira, s.n., collected 22.xii.1989 (ZSS), Windsor Castle, Madagascar; ITS: HQ847463; *trn*L-F: HQ847518. [P051] W. Röösli & R. Hoffman 17/00 (ZSS), Montagne des Francais, Madagascar; ITS: HQ847464; *trn*L-F: HQ847519.

### Funding

Funding for this research was provided by a grant from the Cactus and Succulent Society of America to Dylan Burge, and a grant from the National Science Foundation (DEB 1118783) to Anurag Agrawal. The funders had no role in study design, data collection and analysis, decision to publish, or preparation of the manuscript.

### Grant Disclosures

The following grant information was disclosed by the authors:
Cactus and Succulent Society of America.
National Science Foundation: DEB 1118783.

### Competing Interests

Anurag Agrawal is an Academic Editor for PeerJ.

### Author Contributions

- Dylan O. Burge conceived and designed the experiments, performed the experiments, analyzed the data, contributed reagents/materials/analysis tools, wrote the paper.

- Kaila Mugford conceived and designed the experiments, performed the experiments, analyzed the data, wrote the paper.
- Amy P. Hastings performed the experiments, analyzed the data.
- Anurag A. Agrawal conceived and designed the experiments, contributed reagents/materials/analysis tools, wrote the paper, has expertise in trait evolution.

### DNA Deposition

The following information was supplied regarding the deposition of DNA sequences:
Genbank: HQ847410–HQ847519; JN256214–17; KC189049.

### Supplemental Information

Supplemental information for this article can be found online at http://dx.doi.org/10.7717/peerj.70.

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
