# Peer review of "Phylogeny of the plant genus Pachypodium (Apocynaceae)"

_PeerJ, doi:10.7717/peerj.70_

## Round 0.1 · original submission · Minor Revisions

As you can see below, the reviews are generally positive so I am optimistic that we will be able to publish your contribution once you have addressed the referees' comments. Most comments pertain to specific points that could be made more clear or explicit. But importantly, both Reviewer 1 and 3 questioned the support of your conclusion regarding the monophylly of African/Malagasy Pachypodium. Please address this point carefully in your revisions---weakening your conclusion if needed.

Reviewer 1 ·

Basic reporting

All criteria are met for Basic Reporting, except the layout differs from the recommended 'Standard Sections'. Specifically, the 'Results and Discussion' is separated into 'Results' and 'Discussion', and there is no separate 'Conclusions' section. This is a minor revision to comply with this PeerJ policy.

Figures 2 and 4 appear to be swapped in the reviewer's pdf.

Experimental design

The Experimental Design complies with PeerJ provisions.

There are several minor issues with methods, most dealing with two PeerJ policies 'Methods should be described with sufficient information to be reproducible by another investigator' and 'The investigation must have been conducted rigorously and to a high technical standard'.

The biogeography component becomes a major focus of the figures and discussion, yet there is nothing, specifically, in the methods that suggests how a biogeographical analysis is to be carried out. In particular, the methods used to reconstruct the distributions in the figures are particularly vague and irreproducible. The only hint about how the distributions are reconstructed is in the figure legend: "Data for distributions is approximate, adapted from Lüthy (2006) and Vorster and Vorster (1973)." With the diversity of software, including maxent, for reconstructing ranges, the wide availability of climate data (from, e.g., WorldClim), and the apparent availability of specific collection localities, is there a reason a more rigorous approach to inferring distributions was not used?

There are several minor points that could use clarification.

(1) line 141-142: "To account for this conflict, we ran *all* of our phylogenetic analyses on the separate trnL-F and ITS alignments..", but later it's stated that partitioned and concatenated analyses were carried out. It would appear that not *all* analyses were based on single loci. Perhaps just mention that both concatenated and individual locus analyses were carried on line 142.

(2) line 168: "This strategy checks for shorter trees than those found by the initial search, demonstrating that the final consensus tree reflects all of the most parsimonious trees". I wonder if this conclusion necessarily follows. If I follow these methods, a 'quick-and-dirty' MP analysis limited subsequent, more thorough, analyses to a subset of the tree topology space. However, I'm not convinced that the strict consensus constraint tree based on the quick-and-dirty analysis necessarily will 'reflect' the topology of the true set of MP trees if the true MP tree topologies were simply not found during the relatively shallow quick-and-dirty search.

(3) One of the stated goals is to assess the monophyly of Pachypodium. For the ITS dataset, there is a single outgroup taxon. I wonder how monophyly was assessed with a single outgroup taxon without a priori (and circularly) assuming that Pachypodium is monophyletic and sister to the outgroup taxon. The analyses with trnL-F strongly suggest monophyly, but the analysis was based on a single chloroplast locus. I am not terribly concerned that a single locus was used to infer monophyly, but it would be especially helpful if morphological synapomorphies could be reported for Pachypodium which further support monophyly.

(4) On a related note to (3), why are support values not provided for the outgroup taxa? In Figure 4, in particular, why is there no support listed for the root node of Pachypodium (whereas there is support indicated for this node in the trnL-F tree in Fig. 3)? Does the combined tree not have high support for the monophyly of Pachypodium?

(5) Clarify the discussion concerning the type(s) of consensus trees used. Is the *50%* majority rule (MR) consensus tree used throughout? (The text just says majority rule). Why was the 50% MR tree selected for the Bayesian analysis? Were maximum clade credibility trees or other Bayesian consensus approaches examined? Do the figures present the 50% MR tree? If so, what method was used to estimate the posterior probabilities from the Bayesian analyses for the 50% MR tree?

(6) Frequently (e.g., lines 197, 199, 200) the manuscript states there are X number of resolved nodes. Is the criterion for being 'resolved' based on the 50% MR tree for the Bayesian and MP analyses? I haven't seen this criterion used to assess the level of resolution, and I am not convinced that a 50% consensus value is sufficient support to conclude that a node is resolved.

(7) Technically, the terminals are nodes, so statements that there are X nodes in a given tree aren't technically accurate as they stand. Just change to 'internal nodes'.

(8) On line 140, the manuscript states that conflicts between trees based on different loci would be noted. However, later, the manuscript reports somewhat non-specific comparisons between trees:

line 211 “…support similar relationships to those indicated by maximum parsimony and Bayesian analyses.”

line 218: “Overall, relationships recovered on the basis of trnL-F support those obtained from ITS…”

These two statements are vague. I'd like to see either a more quantitative or a more explicit comparison between trees. The manuscript reports how many nodes are supported in consensus trees based on each of the three alignments, but not whether the same nodes are supported among trees. How many of the supported nodes are present in all trees (trnL-F, ITS, and concatenated)? Are there particularly surprising conflicts among loci?

(9) What criteria were used to define alignment ambiguity and to exclude positions in the (ITS) alignment? The criteria on line 127 are vague: "For ITS, several indel- and repeat-rich regions (54 bp total) were excluded due to alignment ambiguity." and line 184: "...indel- and repeat-rich regions that could not be readily aligned, and were thus excluded from analysis, as described above." These statements do not provide sufficient information for reproducibility. There are multiple computer programs with (reproducible) techniques for assessing homology and that provide quantitative metrics to justify the exclusion of poorly aligned regions, including the program Hmmalign (S. Eddy), which provides a Stockholm-formatted alignment with a posterior probability of 'homology' for each column of an alignment.

Validity of the findings

The main conclusion of the discussion is unfounded based on the data and results presented. The discussion states (line 236) "Our results suggest that neither African nor Malagasy Pachypodium are monophyletic (Fig. 3)." However, there are two broad interpretations that are consistent with the provided results: (1) neither African nor Malagasy Pachypodium are monophyletic, or (2) the data do not provide enough information to draw conclusions concerning the monophyly of the African and Malagasy taxa. It is this second interpretation that would seem to be warranted. Lack of resolution concerning monophyly does not imply that the taxa are polyphyletic. The key nodes are not resolved in the figures (they appear to be soft polytomies). Moreover, the taxon sampling in the trnL-F tree suggests that the African taxa are, in fact, monophyletic, but it would appear (based on the figure) that P. bispinosum has not been sampled for trnL-F, so monophyly of African taxa based on trnL-F is unclear. Based on all the trees presented, there are three mutually exclusive interpretations that are consistent with what is presented: (1) African taxa are paraphyletic and basal, and the Malagasy taxa are uniquely derived from within an African grade, (2) African taxa are monophyletic and sister to a monophyletic Malagasy clade, or (3) neither the Malagasy, nor the African taxa are monophyletic. To distinguish among these hypotheses, perhaps the authors could investigate and apply one of the tests of topology (KW / SH / AU / SOWH tests).

Is the apparent lack of monophyly of, e.g, P. densiflorum, P. brevicaule, and other previously-recognized species surprising? What do you make of this - incomplete lineage sorting, incorrect specimen determinations, or something else?

Additional comments

Burge et al. present the first phylogenetic analyses of the genus Pachypodium, a plant group in which evolution experimented with growth form. They sample all the specific and sub-specific taxa within Pachypodium and assess their evolutionary history using two loci: one nuclear (ITS) and one chloroplast (trnL-F). As a preliminary analysis, the results and data are useful and warrant publication. Most of my comments are minor, but I don't think the data are sufficient to accurately determine the monophyly (or lack thereof) of the Malagasy or African taxa. I don't see this as a major barrier to publication, but the limitations of the data set should be discussed frankly. Additionally, I point out several minor issues that are easily addressed. Issues are discussed in greater depth in other sections of this review.

It is worth reiterating that "Negative / inconclusive results are acceptable" according to PeerJ policies, so an inconclusive result concerning monophyly is still appropriate for publication.

Figures are beautiful.

Reviewer 2 ·

Basic reporting

Overall the work is sound and thorough, and should significantly contribute to the understanding of the Pachypodium phylogeny and biogeography.

There is a minor discrepancy in the stated number of species in the genus Pachypodium. In the introduction, the authors state that the genus consists of 21 species (l. 21). Later in the text (l. 76), they state that the sequenced samples represent 27 taxa, contradicting the earlier statement. For comparison, the Wikipedia article about the genus states that it comprises 25 species.

Experimental design

I found no problems with the experimental design but have a few minor comments to the phylogeny reconstruction in MrBayes:

1. The settings used in MrBayes are not given. It would be helpful to know whether there are rate and/or topology constraints between the two partitions. A short statement that the default settings were used (if this is the case) or a list of parameters that were change would clarify this minor issue.

2. The version of MrBayes used (3.0) dates back to 2005. Newer versions (such as 3.2 from 2011 or the very recent 3.2.1) are free of some bugs present in 3.0 and could potentially give better results.

3. It is not clear from the text which parameters in the MCMC simulation were inspected for convergence.

Validity of the findings

In my opinion the phylogeny reconstruction conforms to the state-of-the-art in the field and thus I believe the findings to be correct.

Additional comments

There are a few minor technical issues with the manuscript:

1. The figure captions don't match the figures.

2. In fig. 2, it might be helpful to mark the geographic distribution of samples, as is done in fig. 3.

3. The abbreviation PP is explained twice in figure captions but not in the main text.

4. The values for PP, MP and ML BS are sometimes given as exact values and sometimes as inequalities (e. g. l. 248 vs. l. 250).

·

Basic reporting

The manuscript adheres to the formatting and content guidelines of PeerJ. I found one persistent error with respect to the figure numbering in the manuscript that made reading challenging at times. Often when it is clear that the authors are referring to, for example, the result of the combined phylogenetic analysis (Fig. 4), they instead cite Fig. 3. This is true of Fig. 2 as well, and it could be a problem with the table references in the manuscript as well. I just wanted to point this out as I think it is easily fixed by careful revision of the manuscript by the authors.

Experimental design

The authors provide a detailed account of the phylogenetic analyses performed on the nuclear ITS and trnLF regions as well as the analyses on the combined data set. A critical element that has been left ouf of these descriptions is the identification and treatment of conflicting phylogenetic signal between the two loci. On lines 136-143 the authors indicate that significant phylogenetic conflict was identified between the two loci and they refer the reader to the section outlining the Phylogenetic analyses for further information on this topic ("see Phylogenetic analyses"). But upon reading this section, I found no information phylogenetic conflict. The only other mention of phylogenetic conflict that I found in the manuscript is a single sentence in the results section: "There were some cases of conflict between the trees based on these two regions (Fig. 2 [should be Fig. 3])." This treatment of phylogenetic conflict is insufficient. Rather than leave the reader to try to compare the two trees in Fig. 3, the authors should at minimum identify the specific taxa or clades that show evidence of phylogenetic conflict and discuss the potential implications of such conflict. Given that the two loci are from plastid and nuclear genomes, such evidence of conflict could have a biological explanation (i.e. hybridization, plastid capture, etc.) Thus, if there is indeed "significant" conflict, I would expect some mention of the patterns of phylogenetic discordance between these two genomes in the discussion section as well. Additionally, in the case of significant conflict, do the authors feel that it is appropriate to simply combine the data without considering the implications of such a decision? Given that this issue is certainly common in phylogenetic studies, some discussion of these issues and various ways of dealing with phylogenetic conflict (e.g. removing offending taxa, etc.) would be appropriate.

Validity of the findings

With one exception, I feel that the principle findings of the manuscript are valid and do not over-reach the signal of the data. The one exception to this is the author's conclusions regarding the monophyly of the African (and Malagasy) species. The phylogenetic analyses presented in this study suffer from a lack of resolution at the base of the tree, such that the relationships between well-supported clades are not resolved. While it is certainly true that this lack of resolution does not support the monophyly of the African and Malagasy species, it also does not support the non-monophyly of these species. Instead, the data lack the phylogenetic signal to support or reject the monophyly of these geographic groups. Furthermore, it appears from Fig. 3 that the trnLF data actually might support (0.97 posterior probability) the monophyly of the African species. In light of all of this, I was surprised that the authors repeatedly stated that the data indicate that the Africa taxa are not monophyletic (lines 241-242 and 270-271). I feel that this conclusion over reaches the signal in the data. But ultimately this can be remedied with some small changes in wording.

Additional comments

Overall, I found the manuscript to be well written and easy to follow. I would suggest perhaps labeling the individual trnLF and ITS trees a bit more completely (i.e. similar to the way the combined tree was labeled). Without any additional labeling it is often difficult to examine the differences between the two trees and the combined tree. I very much liked the addition of the line drawings to the phylogenetic tree figures.

---

## Round 0.2 · Minor Revisions

Thank you for revising your manuscript. Curiously, your response letter provides no answer to some of the points raised by Reviewer 1, and says nothing about Reviewer 2's points. Perhaps you truncated it by mistake?

Please reorganise your response letter such that it addresses *all* referee comments from the previous round, point-by-point and in the original order.

~

---

## Round 0.3 · accepted · Accept

Thank you for thoroughly addressing the referees' comments. I am pleased to accept your manuscript for publication.